# DNAGRINDER: A LIGHTWEIGHT AND HIGH-CAPACITY GENOMIC FOUNDATION MODEL

## ABSTRACT

The task of understanding and interpreting the complex information encoded within genomic sequences remains a grand challenge in biological research and clinical applications. In this context, recent advancements in large language model research have led to the development of both encoder-only and decoder-only foundation models designed to decode intricate information in DNA sequences. However, several issues persist, particularly regarding the efficient management of long-range dependencies inherent in genomic sequences, the effective representation of nucleotide variations, and the considerable computational costs associated with large model architectures and extensive pretraining datasets. Current genomic foundation models often face a critical tradeoff: smaller models with mediocre performance versus larger models with improved performance. To address these challenges, we introduce dnaGrinder, a unique and efficient genomic foundation model. dnaGrinder excels at managing long-range dependencies within genomic sequences while minimizing computational costs without compromising performance. It achieves results that are not just comparable but often superior to leading DNA models such as Nucleotide Transformer and DNABERT-2. Furthermore, dnaGrinder is designed for easy fine-tuning on workstation-grade GPUs, accommodating input lengths exceeding 17,000 tokens. On a single high-performance GPU, it supports sequences longer than 140,000 tokens, making it a highly efficient and accessible tool for both basic biological research and clinical applications.

## 1 INTRODUCTION

Foundation models (aka large language models) such as BERT (Devlin et al., 2019) and GPT (Brown et al., 2020), have demonstrated their stellar performance in learning the complex characteristics and structures of natural languages, making them well-suited for a variety of subsequent applications, such as sentiment analysis, text generation, and translation (OpenAI et al., 2024). These foundation models have recently been adapted to analyze biological sequences as their deep structure and large-scale parameters are well suited for dealing with the intricacy of biological sequences and structures (Ji et al., 2021; Dalla-Torre et al., 2023; Nguyen et al., 2023; Zhou et al., 2024; Outeiral & Deane, 2024; Wang et al., 2024b; Fang et al., 2024; Wang et al., 2024a). Biological sequences composed of nucleotides like DNA and RNA, as well as amino acids forming peptides and proteins, are regarded as natural languages of life and can be effectively leveraged by using the technology of foundation models to uncover the underlying patterns and functions they encode (Benegas et al., 2023). Typically, these foundation models build robust feature representations from biological sequences through a process known as pretraining. Encoder-based models like BERT perform such pretraining by using a method called Masked Language Modeling (MLM), where they predict the actual words of some masked or corrupted ones in given sequences. By pretraining on millions of biological sequences, foundation models gain a comprehensive contextual understanding of the given sequences. Once trained, they only need a few fine-tuning steps to be effectively applicable to specific downstream tasks (Liu et al., 2024), including prediction of epigenetic marks, gene expressions, protein folding structures, and more.

Understanding the genetic and epigenetic regulations encoded in the genomic sequence and their interactions has been a focal research area in genomics. As the technologies of foundation models have advanced, several models designed specifically for DNA sequences and downstream applications have emerged. Current DNA foundation models, such as DNABERT (Ji et al., 2021), DNABERT-2

(Zhou et al., 2024), and Nucleotide Transformer (NT) (Dalla-Torre et al., 2023), are primarily based on encoder architectures, while others like HyenaDNA (Nguyen et al., 2023) adopt a decoder-only framework. These models aim to capitalize on the strengths of transformer architectures, adapting them to the unique challenges of genomic data.

DNABERT (Ji et al., 2021), as a pioneering DNA foundation model, is capable of extracting context-specific feature representations from large quantities of DNA sequences and addressing various genomic-specific prediction tasks. Despite its widespread use in recent years, the original DNABERT faces several technical limitations. Firstly, DNABERT is pretrained exclusively on the human reference genome, which not only ignores genome diversity across different species but also creates repetitions in the dataset. Specifically, although the human reference genome comprises 3 billion base pairs (bp), DNABERT employs data augmentation to increase the dataset size in order to make pretraining effective for building encoder-based models. Nonetheless, the repetitive sequences, in fact, limit the overall effectiveness of pretraining. Second, the use of overlapping k-mer tokenization can cause information leakage between adjacent tokens during pretraining, while non-overlapping k-mer tokenization can significantly alter the content in cases of sequence addition or deletion. Lastly, DNABERT is restricted to sequences of up to 512 tokens during pretraining, which limits its ability to analyze longer sequences in downstream tasks.

DNABERT-2 (Zhou et al., 2024), an advanced model of DNABERT, replaces the k-mer tokenization with Byte Pair Encoding (BPE), a compression algorithm that counts and merges DNA nucleotides based on their frequency. This encoding effectively avoids information leakage in pretraining and reduces the length of tokenized sequences. DNABERT-2 also incorporates improved positional encoding and Attention with Linear Bias (ALiBi) (Press et al., 2021b) to extend sequence length for downstream applications. However, these improvements and extensions are constrained by the original maximal pretraining length of 128 tokens. Additionally, DNABERT-2 applies the GEGLU activation function (Shazeer, 2020) to improve the convergence of the pretraining process. However, this activation function uses two linear layers, resulting in a parameter size similar to the original BERT and, consequently, longer fine-tuning processes for downstream applications.

NT (Dalla-Torre et al., 2023), which also builds upon the BERT architecture, supports longer sequences. However, its first-generation models, with parameters ranging from 500M to 2500M, are considerably larger than the original BERT, leading to higher computational costs for pretraining and fine-tuning. While the second generation of NT attempts to mitigate these issues, it requires a substantially higher number of training tokens ranging from 300B to 900B driven by the Chinchilla scaling laws, resulting in computation-intensive pretraining and fine-tuning processes.

HyenaDNA (Nguyen et al., 2023), being a decoder-only model, benefits from shorter training time due to its implicit convolution layers. However, it falls short in accuracy compared to the other models mentioned above.

In summary, the restriction on the lengths of sequences processed, the large model parameters, and the high computation cost in pretraining and fine-tuning are common critical issues of the existing foundation models for genomics.

In addition to these drawbacks in the existing methods, the selection of pretraining datasets (Sanabria et al., 2023; Nguyen et al., 2024) also plays a crucial role in the model performance. Typically, these models utilize either the human reference genome, multispecies reference genomes, or a human reference genome augmented with specific variant structures. While multispecies data aims to address the issue of limited genome diversity, the mutually exclusive use of these datasets means each model is constrained to learning from a single source.

To address these limitations, we introduce dnaGrinder, a refined genomic foundation model, whose main contributions can be summarized as follows: 1) We incorporate Flash Attention 2 to optimize computational speed during pretraining and inference; 2) We employ Sequence Length Warmup in pretraining to stabilize training and effectively capture features across varying sequence lengths; 3) A memory-efficient BPE tokenizer is designed to improve memory usage while maintaining representative tokenization for long genomic sequences; 4) A novel approach is introduced to expand the pretraining dataset by effectively increasing genome diversity rather than simply adding similar or repetitive sequences. Through extensive experiments on several downstream benchmarks, we demonstrate that dnaGrinder achieves performance exceeding or comparable to state-of-the-art

models while overcoming input length constraints and requiring fewer parameters and less GPU time for both pretraining and fine-tuning.

## 2 METHODS

In this section, we present an overview of dnaGrinder's architecture, detailing its features and enhancements. We also discuss the specific implementation of the pretraining strategies employed to integrate these architectural improvements, providing insights into how these modifications contribute to the model's overall performance and efficiency.

### 2.1 MODEL

The dnaGrinder model employs an encoder-only transformer architecture (Figure 1.a). DNA sequences are first converted into numerical representations using Byte Pair Encoding (BPE) tokenization (Sennrich et al., 2016). These numerical representations are then transformed into sequences of embeddings through an embedding layer. Unlike most encoder-only models that use absolute positional embedding (Devlin et al., 2019) or rotary positional embedding (Su et al., 2024), we utilize Attention with Linear Biases (ALiBi) (Press et al., 2021b), which is introduced at the beginning of the attention computation. To improve computational, memory, and inference efficiency, we employ sequence length warmup (Figure 1.b) (Press et al., 2021a; Li et al., 2022) to the pretraining phase and adopt Flash Attention 2 (Dao, 2023) as our attention mechanism. We also experiment with several architectural enhancements, including the SwiGLU (Shazeer, 2020) activation function and token random replacement (Dalla-Torre et al., 2023). During the pretraining stage, we incorporate dynamic masking (Lan et al., 2020) to improve the model's learning capability.

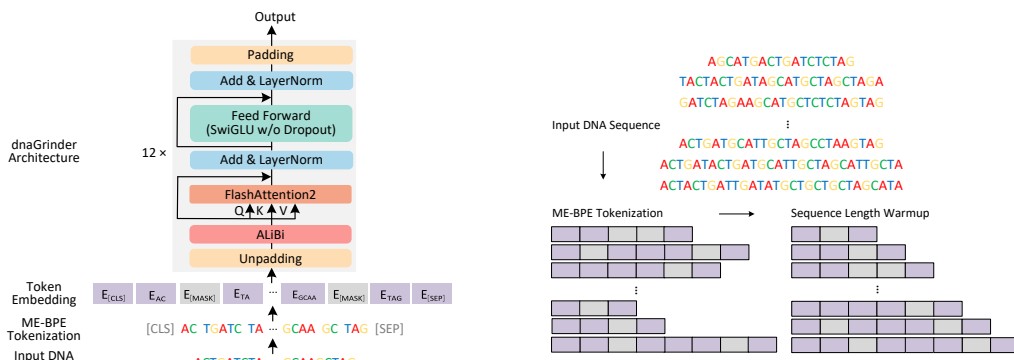

(a) Architecture of dnaGrinder.

(b) Sequence length warmup rearranges the sequences after ME-BPE tokenization.

Figure 1: Sketches of (a) the architecture and (b) characteristics of dnaGrinder.

### 2.1.1 MEMORY-EFFICIENT BPE TOKENIZATION (ME-BPE)

Byte Pair Encoding (BPE) (Sennrich et al., 2016) is a data compression algorithm that segments words by counting the co-occurrence frequency of subwords. For DNA data, BPE starts with a base vocabulary of four characters of bp (A, C, G, and T). In each iteration, it counts the frequency of each consecutive pair of character segments. The most frequent pair is identified and merged into a new subword, effectively reducing the number of distinct pairs. This process is repeated until the vocabulary reaches the desired size, which is 4,096 tokens in the current implementation of the model. By merging frequent pairs, BPE captures common patterns and motifs in the DNA sequences, improving both the efficiency of tokenization and the model's ability to learn meaningful representations. The final vocabulary, therefore, consists of the most frequent and representative tokens derived from the set of given sequences, enlarging the tokenization length and reducing computational complexity.

Counting and merging pairs during BPE tokenization are memory-intensive processes that involve multithreading (Kudo & Richardson, 2018). The memory consumption depends on both the number of sequences and the length of each sequence. For instance, the corpora file of DNABERT-2, where each sequence is 1,000 bp long, and the total size is 30GB, requires over 1TB of memory during training, according to the original authors.

For our model, with each sequence length set to 12,200 bp and a corpus size exceeding 118GB, it is impractical to load the entire corpus into memory at once. Our tests indicate that the maximum manageable corpus size per file for BPE tokenization, with each sequence length set to 12,200 bp, is approximately 20GB, which would still require around 1.8TB of memory. To address this issue, we present ME-BPE, a novel BPE tokenization that split the corpus into smaller files and train iteratively on one file at a time, thereby managing memory usage more effectively.

Our ME-BPE tokenizer begins by processing the first file of sequences to generate an initial vocabulary of 4,096 tokens based on its content. For each subsequent file, the tokenizer updates its vocabulary using the content of the new file. Suppose tokens in the new file appear more frequently than some of the existing tokens in the vocabulary. In that case, the tokenizer replaces the less frequent tokens to maintain a vocabulary size of 4,096. If a token from the new file already exists in the current vocabulary, its frequency will be updated to include the new occurrences. However, if a token was previously excluded from the vocabulary, its earlier frequency is not retained, and only the frequency from the current file is considered. To mitigate the risk of missing some of the most frequent tokens in the entire corpus, the size of each split file plays a critical role. Larger file splits can capture more representative token frequencies across sequences, reducing the likelihood of omitting important tokens. By balancing memory efficiency and file size, this greedy process is necessary to ensure that the vocabulary adapts to many sequences as it processes more files. However it may not necessarily capture all the most frequent tokens, especially for less frequent tokens, in the entire dataset. After processing all files, the final vocabulary will contain up to a predefined number of the most frequent and relevant tokens, which is 4,096 in our current implementation of dnaGrider.

### 2.1.2 SEQUENCE LENGTH WARMUP (SLW)

During pretraining, encoder-based models usually randomly sample data from the entire dataset according to the batch size. This approach works well when the variance of sequence lengths is minimal, as observed in models such as DNABERT (Ji et al., 2021) and NT (Dalla-Torre et al., 2023), which use fixed-length k-mer sequences. On the other hand, with BPE, even if the original sequence length is fixed, the number of tokens after tokenization varies. For instance, DNABERT-2 (Zhou et al., 2024) restricts its pretraining sequences to a maximum of 128 tokens. Notably, despite the use of BPE, the length of DNABERT-2's pretraining sequence varies little, which does not significantly affect its training strategy.

In contrast, our model deals with sequences of 12,000 bp long. Given the use of BPE and a substantial portion of our training sequences from multispecies, the tokenized sequences have over 700 to more than 2300 tokens. If sequences are randomly sampled to form a batch for training, sequence lengths may vary considerably, and we need to pad these sequences to have the longest, uniform length in each batch. Because of the long sequence length resulting from padding, the pretraining is prolonged. Since the sequence lengths vary from one batch to the next, model performance fluctuates across batches. To address these issues, we adopt a sequence length warmup strategy, often used in pretraining decoder models (Press et al., 2021a; Li et al., 2022). This strategy arranges the sequences in increasing order of their number of tokens, which helps to reduce training time and enhance stability as the variance in gradients increases.

In addition, we employ a data augmentation technique akin to that utilized in NT to generate a comprehensive set of training sequences. Initially, we partition the genome of each species into overlapping segments, each with 12,200 bp in length. Each segment is designed to overlap with its predecessor and successor by 100 bp at both the beginning and the end. From these overlapping segments, we extract a 12,000 bp segment. To enhance the diversity of the training set, the starting positions for these extracted segments are randomly selected within the initial 200 bp of the overlapping segments. This extraction process is repeated multiple times, resulting in a training sequence set comprising 300 billion tokens, which is consistent with the quantities typically employed in other genomic foundation models. Although different segments derived from the same genomic region of a species exhibit variability, BPE tokenization ensures that the tokenized sequences

maintain comparable lengths. Finally, the augmented sequences originating from the same genome are organized together in the final pretraining dataset according to their respective sequence lengths.

To the best of our knowledge, our model is the first encoder-based architecture to incorporate SLW in pretraining. By leveraging SLW, we organize the pretraining process by the order of species. Our findings (Section 4) demonstrate that the model is capable of effectively learning sequence features and representations after being trained on a dataset comprising just 69.5 billion tokens.

The following observation can help appreciate SLW's contribution to model performance. Sequences characterized by a greater number of repeated elements (or simple patterns) exhibit lower complexities and reduced entropies, enabling them to be compressed into fewer tokens. This compression results in shorter sequence lengths in terms of token count. In the pretraining phase, sequences with low complexities are prioritized for processing over those with high complexities. The model first acquires the simpler patterns inherent in low-complexity sequences before advancing to the more intricate patterns in high-complexity sequences. This approach of initially focusing on simple patterns facilitates the model's ability to learn complex patterns within more intricate sequences. Consequently, this methodology enhances the overall pretraining process and, in turn, improves the performance of the model.

### 2.1.3 ATTENTION WITH LINEAR BIAS (ALiBi)

When a model is trained on short sequences, such as 512 tokens, its ability to handle longer sequences during inference is known as its extrapolation capability. This presents two challenges: first, the model meets position encodings that are not seen during training; second, the number of tokens processed by the attention mechanism during inference significantly exceeds those encountered during training. Popular approaches, such as Sinusoidal positional embeddings (Vaswani et al., 2017) and Rotary positional embeddings (RoPE) (Su et al., 2024), either impose limitations on the maximum allowed input length or encounter difficulties in maintaining effective attention over long sequences. Specifically, RoPE has been found to have a decaying effect (Xiong et al., 2023), where the model struggles to attend to tokens beyond 4,000-6,000 positions, even with extensive long-context pretraining. This decay in attention scores for distant tokens limits RoPE's effectiveness in handling extremely long input sequences, potentially impacting performance in tasks requiring long-range dependencies. According to the original paper (Press et al., 2021b), ALiBi surpasses T5 Bias and Rotary positional encodings in both training and inference speed, while performing comparably to Sinusoidal encodings.

The ALiBi method is straightforward. It assumes that as the distance between two tokens increases, their association decreases accordingly. Therefore, it penalizes attention scores based on the distance between the two tokens. A pre-defined bias matrix is added to the original attention score computation, which introduces a linear bias to the dot product between the query and key. This bias is an arithmetic sequence with a common difference of 1 and an initial term of $-m(i-1)$:

$$Softmax(q_i K^T + m[-(i-1), \ldots, -2, -1, 0])$$

Our experiments reveal the superior extrapolation ability of ALiBi, particularly in inference tasks like species classification that involve sequences ten times longer than those used during pretraining. Even though the pretraining phase utilized sequences of 12,000 bp, inference tasks were able to extend sequence lengths to 120,000 bp effectively. This aligns with observations in the original study (Press et al., 2021b), where the model's perplexity remained stable as inference token lengths increased.

### 2.1.4 FLASH ATTENTION 2

Flash Attention (Dao et al., 2022) is a fast and efficient vanilla attention enhanced by exploiting IO awareness to compute exact attention scores. Unlike sparse attention methods such as Big Bird (Zaheer et al., 2020) or approximated attention techniques like Linformer (Wang et al., 2020) and Performer (Choromanski et al., 2021), Flash Attention takes advantage of the different capacities and speeds of different memory types in GPUs to accelerate the overall attention computation. For example, SRAM is fast but has limited capacity, whereas High Bandwidth Memory (HBM) offers larger capacity but at slower speeds. By reducing the communication between these memory types, Flash Attention optimizes memory usage and improves computational efficiency.

Unlike the Flash Attention Triton used in DNABERT-2, Flash Attention 2 (Dao, 2023) is twice as fast and optimized for inference, particularly for iterative decoding when the query is a short sequence (e.g., sequence length = 1). This improvement is especially beneficial for our model, as we train on long DNA sequences of 12,000 bp, but DNA sequences in many downstream tasks vary in length, with most not exceeding 1,000 bp. For instance, in DNABERT-2 downstream tasks, all sequences in the GUE dataset are shorter than 1,000 bp.

## 2.2 ARCHITECTURAL ENHANCEMENTS

Beyond improving the primary methods of the model, we have also explored various latest architectural enhancements to optimize model performance. For instance, we experimented with different activation functions and further pretraining.

### 2.2.1 SWIGLU AND GEGLU

DNABERT-2 replaces the ReLU activation function with GEGLU (Shazeer, 2020), a variant of GLU (Dauphin et al., 2016), which has been shown to boost the performance of Transformer models. However, the use of GEGLU increases the parameter size of our model from 63M to 110M due to the two separate linear transformations that the function uses. Specifically, the GELU activation function is applied to the first transformation, and the second transformation serves as a gating mechanism:

$$GEGLU(x, W, V, b, c) = GELU(xW + b) \otimes (xV + c)$$

where $W$ and $V$ are the weight matrices, and $b$ and $c$ are the biases of the transformation. The symbol $\otimes$ represents element-wise multiplication, which modulates the output of the second transformation using the gating signal from the first. This structure leads to a significant increase in the number of parameters, as GEGLU requires separate linear transformations and associated biases for both the gating and output signals, which increases model capacity and computational complexity compared to ReLU and GELU.

SwiGLU (Shazeer, 2020), another variant of GLU, prioritizes parameter size by simplifying the gate computation. To achieve parameter efficiency, SwiGLU utilizes a single linear transformation to compute the gating signal and applies this signal to the result of another linear transformation. This allows SwiGLU to maintain performance while reducing the complexity of the gating mechanism:

$$SwiGLU(x, W, V, b, c, \beta) = Swish_\beta(xW + b) \otimes (xV + c)$$

where $Swish_\beta$ is the Swish activation function with a parameter $\beta$, acting in place of GELU for the gating mechanism. Specifically, for an input dimension $d_{\text{in}}$ and an output dimension $d_{\text{out}}$, GEGLU requires $2 \times (d_{\text{in}} \times d_{\text{out}} + d_{\text{out}})$ parameters. In contrast, SwiGLU achieves the same gating effect with a more parameter-efficient design, requiring only $d_{\text{in}} \times d_{\text{out}} + d_{\text{out}}$ parameters, as the Swish activation allows for a more straightforward gate computation. This makes SwiGLU more parameter-efficient by simplifying gate computation without the additional weights and biases needed by GEGLU. Given the significant increase in parameter size introduced by GEGLU, we opted to use SwiGLU in our model, as it provides comparable performance while substantially reducing the model's overall complexity.

### 2.2.2 FURTHER PRETRAINING

Like DNABERT-2, we also explored further pretraining (Sun et al., 2019) using some downstream datasets. Our model was first pretrained on a general DNA dataset of multispecies reference genomes with human sequences updated with SNP variants. However, downstream classification tasks usually focus on specific regions of the genome, such as genic regions, to predict whether a sequence is a (core) promoter or contains a splicing site. These regions may have some intricate sequence features that the model needs to learn to deliver adequate performance.

Given that our model's pretraining sequences range from 729 to 2314 tokens in length, We employed in-domain further pretraining (Sun et al., 2019), where the model is further pretrained on all downstream datasets, including both the GUE and GUE-plus datasets from DNABERT-2, which contain 10 genomic problems including 36 classification tasks with sequences ranging from 70 to 10,000 bp. This

approach contrasts with the further pretraining of DNABERT-2, which is constrained to only the GUE benchmark consisting of 28 classification tasks due to limitations on its pretraining input sequence length. Another difference from DNABERT-2 is that our model performed 100,000 steps, or about 0.41B tokens, of further pretraining, roughly equivalent to 3-4 epochs on the downstream datasets. In contrast, our model was only further pretrained for one epoch, processing approximately 0.176B tokens across 31,000 steps—about 70% fewer steps and 60% fewer tokens than DNABERT-2's further pretraining.

## 3 DATASETS

### 3.1 PRETRAINING DATASETS

To facilitate effective training of the dnaGrider model, we constructed a comprehensive set of genomic sequences from multiple species, aiming to reduce redundancy while maximizing diversity to capture meaningful genomic variations for robust model training.

#### 3.1.1 THE HUMAN REFERENCE GENOME DATASET

The latest Human Reference Genome (GRCh38.p14) covers approximately 92% of the human genome, encompassing 3.29 billion bp (Nurk et al., 2022). This comprehensive reference includes sequences from all autosomal, sex, and mitochondrial chromosomes. Although the first generation base model of NT replaces the reference genome sequence with 1000 Genome SNP data, introducing alterations to the DNA sequence, it retains 98% redundant content among the samples (Zhang et al., 2024), which limits the NT's ability to learn from the diversity of the human genome.

To mitigate the impact of redundancy, we notice that there are abundant repetitive DNA sequences in genomic sequences. For example, about half of the human genome is repetitive (Treangen & Salzberg, 2012). Such repetitions complicate genomic analyses and mask significant genotypic variations. Therefore, we used the soft-masked assembly sequences from the UCSC Genome Browser to differentiate non-repeats and repeats identified by RepeatMasker (RepeatMasker, 2017) and Tandem Repeats Finder (with a period of 12 bp or less) (Benson, 1999).

To ensure that non-repetitive sequences constitute a substantial portion of each training sequence, we focused on preserving most non-repeating regions while minimizing the inclusion of repeating sequences. To the best of our knowledge, our approach is the first application in the context of genomic models. We initially removed all repetitive regions, focusing solely on the non-repetitive sections. However, we observed that many non-repetitive sequences were short and fragmented. Consequently, we further filtered out non-repetitive sequences shorter than a specified threshold. Following this filtering, we extended the remaining non-repetitive sequences to meet the model's required input length. Even though this extension included a small portion of repetitive regions, we ensured that these sequences were neither fragmented nor redundant. A comprehensive description of the data preparation process can be found in Appendix A.1.1.

#### 3.1.2 1000 GENOME DATA

The 1000 Genome Project dataset contains 3,202 samples, including 2,504 genomes of unrelated individuals and 602 samples from family trios (Byrska-Bishop et al., 2021). These samples originate from 27 geographically structured populations representing African, American, East Asian, and European ancestries. The dataset utilizes the GRCh38.p14 version of the human reference genome as the template. This set of sequence data covers a total of 73,554,796 genetic variants, including filtered Single Nucleotide Variants (SNVs), insertions and deletions (INDELs), and Structured Variants (SVs)—such as large deletions (DELs), insertions (INSs), duplications (DUPs), and inversions (INVs).

To achieve a broader and more diverse data augmentation, we downloaded phased variant call format (VCF) files, where each variant includes one maternal allele and one paternal allele, representing the bp inherited from the respectiv parent. Unlike the NT dataset, which only includes SNVs and INDELs (<50 bp), our dataset also incorporates longer SVs (>50 bp). These SVs (Table 1) represent large-scale genetic alterations in the genome, which can significantly impact gene function and regulation, contributing to genetic diversity and disease susceptibility. Furthermore, to enhance data

Table 1: Total number of phased sites in provided VCFs (chr1-22, chrX)

| Types of variants | Total number of phased sites |
| --- | --- |
| SNVs | 63,993,320 |
| INDELs | 9,459,017 |
| SV-DELs | 54,074 |
| SV-INSs | 32,548 |
| SV-DUPs | 15,234 |
| SV-INVs | 603 |
| Variants | 73,554,796 |

diversity, our training dataset considers both sets of alleles. We utilize maternal and paternal genetic variants (Appendix A.1.2) in our training data, unlike NT, which considers only one set at a time.

### 3.1.3 MULTISPECIES REFERENCE GENOME DATA

The Multispecies Reference Genome dataset includes the reference genomes of 794 species, including a diverse array of organisms such as bacteria, fungi, invertebrates, protozoa, vertebrate mammals, and other vertebrates. We downloaded this dataset directly from NCBI, similar to NT, with the only difference being the exclusion of species with invalid reference links. In processing the data, any character different from a base pair, A, T, C, or G, was transformed into an 'N'. Each DNA chunk was processed to ensure all letters were in uppercase and restricted to bp or N, with any sequence containing 'N' being discarded at the end. This dataset forms the third sequence dataset for pretraining, providing a broad spectrum of genomic data across multiple species for comprehensive genomic studies.

### 3.2 DOWNSTREAM DATASETS

We utilized the GUE dataset from DNABERT-2, which consists of 28 sets of sequences for 7 classification tasks with sequence lengths ranging from 70 to 1,000 bp. The seven genome sequence classification tasks we studied include core promoter detection, promoter detection, transcription factor prediction, and splice site detection for human sequences, transcription factor prediction for mouse sequences, epigenetic marks prediction for yeast sequences, and covid variant classification for virus sequences.

In addition, we incorporated downstream tasks from the NT model. Given that the GUE dataset and NT's downstream tasks overlap in the epigenetic mark prediction, and both datasets include tasks for promoter detection and splice site prediction (albeit with different data), we extended our evaluation to include two additional enhancer-related tasks from the NT model.

To further validate the capability of our model for handling sequences 10 times longer than those used in pretraining, we employed the species classification tasks from HyenaDNA. To ensure consistency and fairness, we selected the same five species used in HyenaDNA: hippo, human, lemur, mouse, and pig. We randomly sampled DNA sequences of 120,000 bp from these species and fine-tuned 6 pretrained models. Our testing revealed that only the HyenaDNA model and our proposed model could accommodate sequences of 120,000 bp long on a single GPU. In contrast, DNABERT-2 and NTs models exceeded the maximum GPU memory capacity even with a batch size of 1.

## 4 RESULTS

### 4.1 BASELINE

We compared dnaGrinder against five top-performing DNA foundation models to assess its performance comprehensively: HyenaDNA, DNABERT-2, NT-500M-1000g, NT-2500M-multi, and NT-50M-multi-V2.

Given that our pretraining data are from multispecies reference genomes and the human genome (version GRCh38) updated with 1000G SNP variants, we included DNABERT-2, NT-2500M-multi, and NT-50M-multi-V2, which were pretrained using multispecies reference genomes. We also included NT-500M-1000g, which was pretrained on the human GRCh38 genome sequences updated with 1000G SNP variants to align with our dataset.

The inclusion of NT-50M-multi-V2 in our comparison was motivated by the fact that it is representative of the second-generation NT models. It incorporates enhancements such as rotary positional encoding, the SwiGLU activation function, and the removal of MLP biases and dropout mechanisms—similar features are used in our model. To the best of our knowledge, this is the first study to compare a model like ours with the second-generation NT.

Additionally, we included HyenaDNA in the comparison because it is a decoder-only model and employs a similar sequence length warmup strategy to ours during pretraining.

## 4.2 SETUP AND METRIC

We assessed the models based on two criteria: computational efficiency and performance on downstream tasks. For computational efficiency, we compared the relative Floating-Point Operations (FLOPs)—the sum of multiplication and addition operations performed during a forward pass. FLOPs were calculated using the H3 dataset from the yeast epigenetic marks prediction task, with sequences of 500 bp long. To assess performance, we used two metrics: Accuracy on 20 application tasks and Matthews Correlation Coefficient (MCC) specifically for the 10 yeast epigenetic marks prediction tasks, for a total of 30 tasks evaluated. Since the DNABERT-2 plus models were not publicly released, using MCC allows us to directly compare dnaGrinder's performance with the reported performance of DNABERT-2 plus in (Zhou et al., 2024) (Table 6). This combination of metrics enables a comprehensive evaluation of each model's computational efficiency and task-specific performance.

## 4.3 RESULTS ON THE GUE BENCHMARK AND ENHANCER TASKS

Table 2 summarizes the performance of six models compared by five evaluation metrics. Notably, dnaGrinder secured the top position in 11 tasks and ranked second in 12 tasks out of 30, achieving the highest overall performance among the six models evaluated. dnaGrinder outperforms the largest, state-of-the-art model NT-2500M-multi in the number of the top-2 tasks and outperforms the second state-of-the-art model DNABERT-2 in the average scores, while significantly surpassing other baselines. This demonstrates dnaGrinder's exceptional efficiency and scalability in genomic sequence modeling without compromising performance.

Table 2: The table presents the performance statistics of 6 models, including the number of parameters, relative FLOPs compared to dnaGrinder, tokens used in pretraining, the top-2 rankings across models (1st ‖ 2nd), and the average evaluation scores on DNABERT-2 and NT downstream tasks. We used the calflops package to calculate the FLOPs for each model; however, during the calculation of HyenaDNA, we encountered a "tensors not on the same device" error, denoted by "/". ↓ indicates that a lower value is better, and ↑ indicates that a higher value is better.

| Model | Params↓ | FLOPs↓ | Trn. Tokens | Num. Top-2↑ | Ave. Scores↑ |
|---|---|---|---|---|---|
| HyenaDNA(1K) | 1.6M | / | 3B | 0 ‖ 0 | 56.99 |
| DNABERT-2 | 117.0M | 1.8 | 262B | 7 ‖ 7 | 70.86 |
| NT-500M-1000g | 480.0M | 5.6 | 50B | 0 ‖ 1 | 64.25 |
| NT-2500M-multi | 2537.0M | 29.4 | 300B | 8 ‖ 4 | 68.32 |
| NT-50M-multi-V2 | 56.0M | 0.6 | 300B | 4 ‖ 6 | 67.70 |
| dnaGrinder | 63.6M | 1.0 | 69B | **11 ‖ 12** | **73.01** |

Among the total 28 GUE benchmark problems, dnaGrinder achieved the best or second-best results in 21 tasks, ranked the best among all methods evaluated (Table 3). The dominance of dnaGrinder over

other baselines is particularly notable in the human and mouse transcription factor prediction problems, reaching the highest or second-highest ACC scores in all ten tasks. Furthermore, dnaGrinder also achieved the highest or second-highest MCC prediction scores in 8 out of 10 tasks, showing strong performance on epigenetic marks prediction tasks. Despite only reaching the second-highest ACC score in 2 out of 6 core promoter and promoter detection tasks, dnaGrinder is ˜40 times fewer in parameters and runs ˜29 times fewer FLOPs when compared with NT-2500M-multi that achieved the highest ACC scores in 4 out of 6 core promoter and promoter detections tasks. This result indicated that dnaGrinder offered a favorable tradeoff between FLOPs, parameters, and model prediction tasks for promoter-related problems.

While dnaGrinder performs similarly to DNABERT-2 and NT-50M-multi-V2 on the enhancer prediction task (Table 4), it achieved an accuracy of 68.50 in the enhancer type prediction task, surpassing the second-best model, NT-50M-multi-V2, by 4.75 points. This result showed dnaGrinder's superior ability to distinguish between different enhancer types, highlighting its robustness in handling more complex genomic classification tasks.

Compared to other baseline models, dnaGrinder also achieved the highest prediction scores in reference to the model parameter size and the number of FLOPs. Although requiring ˜40% more FLOPs and ˜14% more model parameters, dnaGrinder outperformed HyenaDNA in all 30 datasets (Tables 3 and 4) and the species classification task. The large NT-2500M-multi model came out second among the 30 tasks compared, with its performance closely comparable with dnaGrinder (Table 2). However, dnaGrinder is ˜40 times smaller in parameters and runs ˜29 times fewer FLOPs than NT-2500M-multi. Notably, in the species classification task (Table 5), only dnaGrinder and HyenaDNA (160K) successfully handled such long sequences on a single GPU, with dnaGrinder achieving a perfect classification accuracy of 100%, while HyenaDNA (160K) achieving a score of 64.22%. In contrast, models like DNABERT-2, NT-500M-1000g, NT-2500M-multi, and NT-50M-multi-V2 could not process sequences of this length, even with a batch size of 1 on a single GPU. To further illustrate dnaGrinder's extrapolation capability, we conducted GPU evaluations (Table 7) to determine the maximum token length it could handle across different GPUs.

### 4.4 RESULTS OF FURTHER PRETRAINING

Since DNABERT-2 plus, the further pretrained version, was not available for testing, we compared our model's performance with DNABERT-2 plus by using the MCC values for yeast epigenetic marks prediction tasks reported in the DNABERT-2 paper. We then calculated the MCC performance of our model and NT-v2-50M on these 10 classification tasks. The results (Table 6) show that even though our model was trained with fewer steps, it outperformed DNABERT-2 plus on half of the 10 tasks, achieving state-of-the-art performance on these tasks.

## 5 CONCLUSION

We presented dnaGrinder, an efficient and lightweight DNA foundation model that has a high capacity for processing long genomic sequences. We trained the dnaGrinder model on reference genomes of multispecies and human genomes reconstructed from a collection of datasets of SNP variants. In dnaGrinder, we introduced an improved BPE algorithm that significantly reduced memory requirements. Our use of sequence length warmup is the first implementation of this technique in an encoder-based model that accelerates pretraining by aligning with the varying tokenized sequence lengths and our multispecies dataset, a goal that K-mers tokenization cannot achieve. Furthermore, we incorporate several cutting-edge techniques into our encoder-based model, including the ALiBi positional bias mechanism, the SwiGLU activation function, Flash Attention 2, and the elimination of dropout, to significantly improve the overall efficiency and performance of the model. Through these enhanced pretraining strategies and model improvements, dnaGrinder addresses several serious drawbacks in the existing models, such as short pretraining sequence lengths, extensive pretraining datasets, unnecessarily long pretraining processes, and large parameter sizes. dnaGrinder offers a lightweight alternative with a smaller parameter size, reduced pretraining dataset requirements, faster pretraining and fine-tuning times, and, importantly, superior or comparable performance on several genomic applications.

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

# A APPENDIX

## A.1 DATA PREPARATION

To enhance the model's ability to learn from diverse genomic data, we aimed to minimize redundancy by removing repetitive DNA sequences, which can impede the identification of key genomic features. These repetitive regions occupy a significant portion of the genome but offer little benefit to training, as they largely consist of duplicated content that lacks diversity. Our objective was to filter out these repetitive elements while preserving the most informative non-repetitive regions, ensuring that the input sequences were both relevant and met the necessary length for effective training.

### A.1.1 REPEATED AND NON-REPEATED CONTENT

Genomes across species contain repetitive sequences that are present multiple times within chromosomes. These repetitions, ranging from simple patterns like "CGCGCG" to more complex structures, can be categorized into different types, such as tandem repeats or interspersed repeats. Repeats give rise to redundancy and affect genome alignment and assembly, particularly during model pretraining, as they provide duplicated training tokens and position information. In other words, repeated sequences provide little information but incur extra computational burden to pretraining. Therefore, it is necessary to remove these repeats to focus on the unique and informative regions of the genome.

In processing the Human Reference Genome, we utilized the soft-masked assembly data. Initially, we removed all repeat regions, retaining only the non-repetitive sections and documenting their start and end positions. Upon analyzing these sequences, we observed that the majority were short and fragmented, with very few meeting the required input length for our model. To address potential issues associated with variations in input lengths, we implemented the following schemes:

1. **Filtering Short Sequences**: We excluded non-repetitive sequences shorter than a specified length, which varied across chromosomes depending on the proportion of retained sequences relative to the total chromosome lengths. For instance, with a target sequence length of 12,200 bp, non-repetitive sequences shorter than 1,150 bp on chromosome 1 of the human genome were filtered out. This approach ensured that during the subsequent sequence extension phase, we avoided scenarios where short non-repetitive sequences constituted only a small fraction of the final sequence, thus avoiding excessive redundancy.

2. **Extension**:

    (a) **Rightward Extension**: We started the process at position 0 on a selected chromosome, identifying the rightmost index of the first valid non-repetitive sequence. If this sequence was shorter than 12,200 bp, it was then extended to the right along the chromosome until it was 12,200 bp long. If this extension included one or more non-repetitive sequences, the subsequent operation began from the next non-repetitive sequence to the right that had not yet been included. Sequences exceeding 12,200 bp were split to ensure that each segment adhered to this length requirement.

    (b) **Handling 'N' Characters**: In cases where an 'N' (representing unidentified bases) was encountered during rightward extension, the extension was halted, and the sequence was extended to the left to reach the required length of 12,200 bp. Given that 'N' constitutes only 5% of the human reference genome, such unidentified bp rarely appear on each chromosome. We did not observe any cases where the presence of 'N' prevented reaching the target length of 12,200 bp.

After filtering and extension, the final retained set of sequences on chromosome 1 was equivalent to 50% of the original content, reflecting the proportion of non-repeated sequences on this chromosome. Although our data included some repeated sequences, we avoided fragmentation and redundancy. Our approach ensured that non-repeated content formed a substantial part of each training sequence, maximizing the inclusion of meaningful, non-redundant genomic data. The proportions of retained content per human autosomal chromosome plus the X chromosome, were as follows: [0.50, 0.50, 0.55, 0.53, 0.52, 0.51, 0.54, 0.56, 0.47, 0.56, 0.53, 0.54, 0.46, 0.44, 0.44, 0.49, 0.52, 0.51, 0.55, 0.56, 0.50, 0.57, 0.50], which was first introduced in (Treangen & Salzberg, 2012).

In addition, considering that repetitive regions can also contain regulatory elements or genetic variants, we incorporated the complete human reference genome within the multispecies dataset. This inclusion was intended to fill potential gaps in the training data by providing the model with a thorough representation of human genomic features. Despite this inclusion, the proportion of the human genome constituted only 2.7% of the whole multispecies dataset, thereby minimizing the risk of excessive repetition while ensuring that the model benefits from a broad spectrum of genomic information. This approach maintains a balance between leveraging the richness of human genomic data and preventing undue repetition in the training set.

### A.1.2 PARENTAL GENETIC VARIANTS LOCUS REPLACEMENT

Upon obtaining human reference genome sequences of length 12,200 bp, we constructed the final pre-training set by extracting sequences of 12,000 bp long from these sequences with their starting positions randomly chosen from the first 0 to 199 bp.

Subsequently, we randomly selected an individual from the 3,202 samples of the 1,000 Genomes dataset. We then identified all SNVs, INDELs, and SVs (Figure 2.a) of this individual that fell within this extracted 12,000 bp sequence. We replaced these variants at their corresponding positions on this extracted sequence (Kosugi & Terao, 2024). In this replacement, because INDELs (<50 bp) and SVs (>50 bp) are variants of varying lengths, the final length of each 12,000 bp sequence will be different, especially considering that the start index is randomly selected from the first 200 bp. This approach achieves data augmentation by ensuring that the sequences vary significantly.

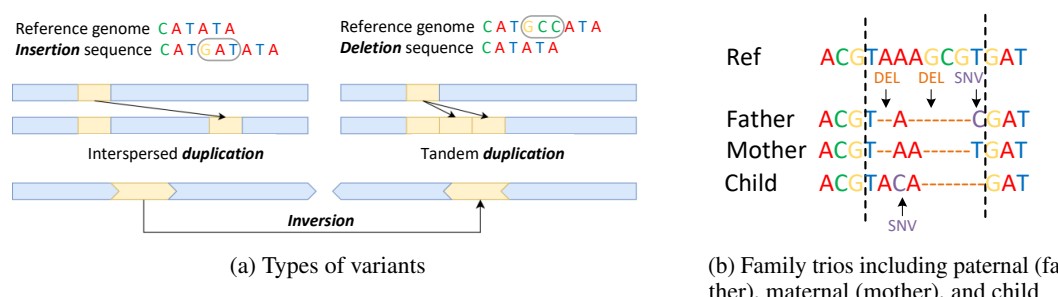

(a) Types of variants

(b) Family trios including paternal (father), maternal (mother), and child

Figure 2: Variants and family trios.

In contrast to the NT-500M-1000g model, which covers only SNVs and INDELs from just one parental lineage (either maternal or paternal, a detail not clarified in their paper), our approach incorporates variants from both maternal and paternal origins (Figure 2). In other words, each extracted 12,000 bp sequence includes two parallel sequences of the maternal and paternal variants. This dual consideration is essential because 18.8% of the sequences of the 1000 Genome project are from family trios, and genetic variations from both parents contribute to the individual's overall genetic makeup. By including variants from both maternal and paternal origins, we aim to capture a more comprehensive representation of genetic variability and enhance the model's ability to account for inherited genetic differences.

## A.2 ALL EXPERIMENT RESULTS

Table 3: The performance of selected 6 models on the GUE datasets.

| Metric: MCC | Epigenetic Marks Prediction | | | | |
|---|---|---|---|---|---|
| | H3K79me3 | H3K14ac | H3K9ac | H4 | H4ac |
| HyenaDNA(1K) | 54.09 | 31.98 | 50.84 | 73.69 | 38.44 |
| DNABERT-2 | 67.39 | 52.57 | 55.63 | 80.71 | 50.43 |
| NT-500M-1000g | 59.33 | 39.37 | 49.29 | 76.29 | 36.79 |
| NT-2500M-multi | 64.70 | 56.20 | 56.01 | **81.67** | 49.13 |
| NT-50M-multi-V2 | 55.25 | **64.89** | **63.78** | 74.65 | 45.03 |
| dnaGrinder | **67.58** | 57.36 | 59.95 | 81.01 | **54.65** |

| Metric: MCC | Epigenetic Marks Prediction | | | | |
|---|---|---|---|---|---|
| | H3 | H3K36me3 | H3K4me1 | H3K4me2 | H3K4me3 |
| HyenaDNA(1K) | 67.17 | 48.27 | 35.83 | 25.81 | 23.15 |
| DNABERT-2 | 78.27 | 56.88 | 50.52 | 31.13 | 36.27 |
| NT-500M-1000g | 72.52 | 45.58 | 40.45 | 31.05 | 26.16 |
| NT-2500M-multi | 78.77 | **61.99** | **55.30** | 36.49 | 40.34 |
| NT-50M-multi-V2 | 69.80 | 52.66 | 39.46 | 27.76 | 41.4 |
| dnaGrinder | **80.23** | 56.65 | 47.22 | **45.22** | **48.03** |

| Metric: ACC | Core Promoter Detection | | | Promoter Detection | | |
|---|---|---|---|---|---|---|
| | all | notata | tata | all | notata | tata |
| HyenaDNA(1K) | 76.64 | 79.46 | 72.26 | 88.76 | 93.34 | 76.18 |
| DNABERT-2 | 81.77 | 82.38 | 84.50 | 93.07 | 96.74 | 83.19 |
| NT-500M-1000g | 81.87 | 82.43 | 83.03 | 92.88 | 95.36 | 87.60 |
| NT-2500M-multi | **84.03** | **83.38** | 83.84 | **95.30** | **97.11** | 82.30 |
| NT-50M-multi-V2 | 83.47 | 81.27 | **89.07** | 93.59 | 95.70 | **93.63** |
| dnaGrinder | 82.15 | 83.31 | 87.11 | 92.69 | 96.06 | 83.52 |

| Metric: ACC | Transcription Factor Prediction (Human) | | | | | Splice |
|---|---|---|---|---|---|---|
| | 0 | 1 | 2 | 3 | 4 | reconstructed |
| HyenaDNA(1K) | 79.3 | 80.7 | 70.1 | 66.2 | 77.3 | 62.84 |
| DNABERT-2 | 82.1 | 83.3 | **82.3** | 77.2 | **87.7** | **91.49** |
| NT-500M-1000g | 82.4 | 84.1 | 75.7 | 72.5 | 81.4 | 87.52 |
| NT-2500M-multi | 83.3 | 85.1 | 77.4 | 75.1 | 81.6 | 88.75 |
| NT-50M-multi-V2 | 79.9 | 80.5 | 75.1 | 67.4 | 81.1 | 90.31 |
| dnaGrinder | **85.4** | **86.6** | 80.1 | **77.3** | 85.6 | 89.30 |

| Metric: ACC | Transcription Factor Prediction (Mouse) | | | | | Virus |
|---|---|---|---|---|---|---|
| | 0 | 1 | 2 | 3 | 4 | Covid |
| HyenaDNA(1K) | 51.97 | 85.29 | 82.01 | 57.74 | 60.06 | 14.28 |
| DNABERT-2 | **80.00** | 90.86 | **92.07** | 86.61 | **73.60** | 69.19 |
| NT-500M-1000g | 67.90 | 87.57 | 82.62 | 61.08 | 66.70 | 37.36 |
| NT-2500M-multi | 55.68 | **91.91** | 83.46 | 57.32 | 62.26 | 38.01 |
| NT-50M-multi-V2 | 71.11 | 87.79 | 85.36 | 68.61 | 64.84 | 50.96 |
| dnaGrinder | 74.32 | 91.09 | 90.85 | **88.28** | 69.25 | **69.95** |

Table 4: The performance of selected 6 models on the enhancer prediction tasks from NT.

| Metric: ACC | Enhancer Prediction | |
|---|---|---|
| | Enhancer | Enhancer Types |
| HyenaDNA(1K) | 71.75 | 62.75 |
| DNABERT-2 | **79.25** | 56.50 |
| NT-500M-1000g | 77.00 | 58.50 |
| NT-2500M-multi | 71.75 | 57.75 |
| NT-50M-multi-V2 | 79.00 | 63.75 |
| dnaGrinder | 79.00 | **68.50** |

Table 5: The performance of selected six models on a long sequence classification task on a single GPU. Only dnaGrinder and HyenaDNA can handle such long sequences.

| | Species Classification |
|---|---|
| | 120K Base Pair Length |
| HyenaDNA(160K) | 64.22 |
| DNABERT-2 | / |
| NT-500M-1000g | / |
| NT-2500M-multi | / |
| NT-50M-multi-V2 | / |
| dnaGrinder | **100.00** |

### A.3 COMPARISON WITH DNABERT-2 WITH FURTHER PRETRAINING

To ensure a comprehensive comparison, we also evaluate the performance of dnaGrinder with further pretraining against DNABERT-2 with further pretraining on epigenetic marks prediction tasks (Table 6). Since DNABERT-2 with further pretraining has not been officially released, we relied on the MCC scores reported in the DNABERT-2 paper. Notably, even though our model was pretrained on only one-third of the data used for DNABERT-2's further pretraining, it delivered comparable performance across 10 tasks.

After further pretraining, dnaGrinder experienced a decline in MCC scores for 6 out of the 10 yeast epigenetic mark prediction tasks, with an average decrease of 0.8783. In contrast, the scores improved for 4 tasks, with an average increase of 0.755. These results suggest that further pretraining did not yield significant benefits for dnaGrinder in these tasks. This indicates that while further pretraining might offer some improvements in specific cases, its overall impact on dnaGrinder's performance is limited, and the gains do not substantially enhance the model's effectiveness in this context.

### A.4 GPU MEMORY EVALUATION

To evaluate dnaGrinder's efficiency in compressing nucleotide sequences and handling lengthy input lengths, we randomly selected 11 DNA sequences from each chromosome of the human reference genome. The results (Table 7) illustrate dnaGrinder's capability to effectively handle lengthy genomic sequences, with our ME-BPE tokenization encoding approximately 5 bp per token, even with GPUs that have limited memory. Specifically, dnaGrinder can process sequences of over 17,000 tokens on workstation-grade GPUs with 12GB of memory, such as the RTX 4070. In contrast, high-performance GPUs with 80GB of memory, like the H100 or A800, dnaGrinde can handle sequences exceeding 140,000 tokens. By efficiently managing very long sequences while maintaining a small parameter size and requiring minimal fine-tuning time, dnaGrinder proves to be a highly effective tool for addressing complex genomic challenges, even under resource-constrained conditions.

Table 6: The performance of DNABERT-2 plus (with further pretraining) and dnaGrinder plus (with further pretraining) on epigenetic marks prediction.

| Metric: ACC | H3 | H3K14ac | H3K36me3 | H3K4me1 | H3K4me2 |
|---|---|---|---|---|---|
| DNABERT-2 plus | **80.17** | **57.42** | **61.90** | **53.00** | 39.89 |
| dnaGrinder plus | 78.91 | 56.44 | 56.93 | 46.47 | **45.07** |

| Metric: ACC | H3K4me3 | H3K79me3 | H3K9ac | H4 | H4ac |
|---|---|---|---|---|---|
| DNABERT-2 plus | 41.20 | 65.46 | 57.07 | **81.86** | 50.35 |
| dnaGrinder plus | **49.90** | **67.99** | **60.41** | 80.58 | **52.95** |

Table 7: Sequence lengths in tokens for varying original DNA sequence lengths.

| DNA Sequence Length (bp) | Longest Tokenized Length (tokens) | Shortest Tokenized Length (tokens) |
|---|---|---|
| 120,000 | 24,709 | 21,036 |
| 250,000 | 51,588 | 44,784 |
| 300,000 | 62,137 | 53,209 |
| 500,000 | 103,285 | 89,779 |
| 700,000 | 144,653 | 126,738 |

## A.5 PRETRAINING AND FINE-TUNING INSIGHTS

During an early pretraining trial, we initially planned to pretrain exclusively on the 1000G SNP variant data. However, we discovered that the model was only able to learn features from a single chromosome. For instance, after achieving 60% accuracy on chromosome 1, the model performed poorly on other chromosomes with the same MLM task. This indicated that the model was not learning generalizable data distributions beyond one chromosome. We initially suspected that the data might be too limited for generalization, so we expanded our pretraining data to include SNP variants from chromosomes 1, 21, and 22. Yet, the model still failed to achieve generalization. We concluded that SNP variants, being often physically separated from one another by arbitrary distances on one chromosome, are not suitable for modeling biological data distributions. Consequently, we decided to use complete DNA sequences with SNP variants incorporated.

In another early pretraining trial, we also experimented with dilated attention (Ding et al., 2023) to extend sequence lengths. Although dilated attention allowed us to scale sequences up to 400,000 bp, it was challenging for the model to effectively learn data features. This led to consistently low MLM accuracy that was insufficient for downstream tasks. The poor generalizability of this model can be attributed to the missing information in the model because dilated attention approximates the authentic attention mechanism.

During fine-tuning, we observed that BERT models were quite sensitive to learning rates. Small variations, such as a change of $0.1 \times 10^{-5}$ in the learning rate, could lead to significantly different test results. As a result, we tested our model in a range of learning rates between $1.0 \times 10^{-5}$ and $3.0 \times 10^{-5}$ for most tasks to identify the optimal rate. Additionally, we used five different random seeds for each downstream task, resulting in approximately 100 runs per task to determine the best test results.

For the Covid variant classification of the virus, our tests show that all six models struggle to converge on this dataset. However, dnaGrinder and DNABERT-2 can converge in one or two runs, while the other models have difficulty converging even after multiple attempts. For instance, despite trying ten different random seeds, HyenaDNA consistently failed to converge on this task. Additionally, fine-tuning NT-2500M-multi and NT-500M-1000g using LoRA for five epochs required approximately 700 minutes and 150 minutes, respectively. This significantly increases the time cost for each new attempt with a different random seed. The authors of DNABERT-2 attribute these difficulties to early

convergence to local minima, likely stemming from the substantial mismatch between the pretraining and evaluation data distributions.

For the NT 2.5b-MS model, our tests revealed that its large size and depth led to an early and easy convergence to local minima in some tasks. This resulted in the model struggling to learn data features, with accuracy stagnating around 50%. Moreover, the model's runtime per epoch was significantly longer than that of other models. Even with the use of LoRA, with only about 0.1% of the parameters being fine-tuned, extensive testing with different random seeds was required to surpass 50% accuracy. This explains why the NT 2.5b-MS performed worse in our tests compared to DNABERT-2.

## A.6 IMPLEMENTATION DETAILS

### A.6.1 SPECIES CLASSIFICATION

To ensure a fair comparison, we selected the same 5 species as used in the HyenaDNA paper: hippo, human, lemur, mouse, and pig. For each species, we randomly sampled 11 DNA sequences of 120,000 bp from the reference genome of each chromosome, with 10 sequences allocated for training and 1 for testing. Sequences of any chromosome shorter than 120,000 bp were excluded from the sampling process. The completed dataset includes 1,090 sequences for training and 109 sequences for testing. Among the 6 tested models, only dnaGrinder and HyenaDNA were able to process sequences of this length as input. In contrast, other models were unable to handle the sequence length even with a batch size of 1 on a single GPU.

### A.6.2 PRETRAINING IMPLEMENTATION

We pretrain dnaGrinder on 8 H100 GPUs using MLM with a 15% mask ratio and dynamic masking for each sequence. We use a batch size of 256 and a maximum sequence length of 2314. We train the model for 119,000 steps using the AdamW optimizer with $\beta_1 = 0.9$, $\beta_2 = 0.98$, $\epsilon = 1 \times 10^{-6}$, and a weight decay of $1 \times 10^{-5}$. The learning rate linearly increases from 0 to $4 \times 10^{-4}$ during the first 16,000 steps, followed by cosine annealing for the remaining steps.

### A.6.3 FURTHER PRETRAINING IMPLEMENTATION

We further pretrained dnaGrinder on a single A800 GPU, using MLM with a 15% mask ratio and dynamic masking for each sequence. We use a batch size of 32 and a maximum sequence length of 2241. We train the model for 31,000 steps using the AdamW optimizer with $\beta_1 = 0.9$, $\beta_2 = 0.98$, $\epsilon = 1 \times 10^{-6}$, and a weight decay of $1 \times 10^{-5}$. The learning rate is set to $5 \times 10^{-5}$.

### A.6.4 FINE-TUNING IMPLEMENTATION

For fine-tuning, we applied a consistent architecture across all models, which includes two linear layers. The structure is as follows: the first linear layer is followed by layer normalization, a GELU activation function, and dropout with a rate of 0.1. The output is subsequently fed into a second linear layer, which produces the final classification output.

HyenaDNA, as an exception, utilizes only a single linear layer since it has already integrated the classification layer directly into the model.

In the case of dnaGrinder, we explored 20 different learning rates for each task, including ranges such as $1 \times 10^{-5}$ to $3 \times 10^{-5}$, $3 \times 10^{-5}$ to $5 \times 10^{-5}$, and $5 \times 10^{-5}$ to $7 \times 10^{-5}$. Additionally, we experimented with 5 different random seeds, resulting in a total of 100 model runs per downstream task to identify the optimal test results.

For DNABERT-2 and HyenaDNA, we adopted a learning rate of $3 \times 10^{-5}$, as reported in the DNABERT-2 paper. For the three models of NT, we used a learning rate of $1 \times 10^{-4}$, consistent with the values reported in the original paper. The number of epochs for each task is displayed in Table 8.

Table 8: The number of training steps used for the following tasks: epigenetic marks prediction (EMP), transcription factor prediction on the human genome and the mouse genome (TF-H and TF-M), tata dataset of promoter detection (PD-tata), notata and all datasets of promoter detection (PD-o), tata dataset of core promoter detection (CPD-tata), notata and all datasets of core promoter detection (CPD-o), splice site prediction (SSP), covid variant classification (CVC), enhancer and enhancer types (enhancer), and multi-species classification (species).

|  | EMP | TF-M | TF-H | PD-data | PD-o | CPD-data | CPD-o | SSP | CVC | Enhancer | Species |
|---|---|---|---|---|---|---|---|---|---|---|---|
| **Epochs** | 5 | 10 | 5 | 10 | 5 | 10 | 5 | 5 | 5 | 5 | 5 |

## A.7 PRETRAINING DATA AVAILABILITY

For the softmask assembly of the human reference genome, we have selected the GRCh38 version from the UCSC browse: `https://hgdownload.soe.ucsc.edu/goldenPath/hg38/bigZips/`. For the 1000 Genome variants, we have selected the 1000 Genome project: `https://ftp.1000genomes.ebi.ac.uk/vol1/ftp/data_collections/1000G_2504_high_coverage/working/20220422_3202_phased_SNV_INDEL_SV/`. For the multi-species reference genome, we have selected from NCBI: `https://www.ncbi.nlm.nih.gov/`.

