# OpenReview forum: "dnaGrinder: a lightweight and high-capacity genomic foundation model"
_ICLR.cc/2025/Conference — ICLR 2025 Conference Withdrawn Submission_

### Official Review · Reviewer_9FMg · 2024-10-30

**Soundness:** 3
**Presentation:** 3
**Contribution:** 2
**Rating:** 5
**Confidence:** 4

**Summary:**

In this paper, the authors propose and implement a genomic foundation model (dnaGrinder) that overcomes two major limitations of existing models: limited genomic context and inefficient training and fine-tuning. To achieve this, they adapt existing techniques, such as Flash Attention 2, Sequence Length Warmup, and a new memory-efficient BPE tokenizer. They then compare their foundation model's performance across a large set of benchmarks against other models.

**Strengths:**

- The authors have conducted an extensive survey of methods and strategies to improve training and inference efficiency, and it has paid off.
- The proposed method has been compared to existing methods across a large set of benchmarks.
- More complex forms of genomic variation (e.g., structural variations and trio datasets) are included in the training set.

**Weaknesses:**

**Major concerns**
- The biggest question is how well this foundation model performs its primary function: predicting masked tokens. It seems possible that these models are simply memorizing the genome, which, relatively speaking, is not that large. The authors mention that (line 1,000), “For instance, after achieving 60% accuracy on chromosome 1, the model performed poorly on other chromosomes with the same MLM task.” Carefully assessing the model's performance on held-out data, at least on held-out chromosomes, is key. Although there are many homologous sequences across chromosomes, this approach is a good start.
- In reviewing the performance metrics (e.g., Table 3), it’s unclear how meaningful these differences are or whether they will yield any practical biological outcomes. There are numerous new techniques, architectures, and paradigms that improve foundation models in one way or another. A clear objective is necessary for developing a foundation model rather than simply applying different “module” combinations.
- Comparing performance metrics between two methods cannot rely solely on point estimates. Providing a measure of uncertainty, such as confidence intervals or standard deviations, would help. Since all these methods run on the same test datasets, using paired tests like paired bootstrapping would be even better.
- There’s some uncertainty around excluding more than half of the genome (repeats). Although many repeats are simple, they are vastly heterogeneous; many are actively silenced and regulated and play crucial roles in development and disease.

**Minor Points**
- The sentence on line 49 requires citations for each application.
- In Fig. 1b, it would be helpful if sequences before tokenization were numbered or labeled, or if arrows were added to indicate that sequences are reordered, not padded.
- Lines 326–330 are unclear: Is the proposed model further pre-trained using 0.41B tokens or 0.176B tokens?

**Questions:**

- How and why does the large context window improve model performance?
- The in-memory tokenization scheme is a greedy approach, and it’s unclear how suboptimal this tokenization may be. Would it be possible to perform tokenization offline, with multiple passes over the input sequences? Though this might be slower and less efficient, it would be done only once, and given its potential impact on training and inference performance, it might be worth the extra effort.
- How does the model perform when predicting masked tokens in repeat regions?
It’s unclear how beneficial the addition of more complex genomic variations is — an ablation study could help clarify this, especially since all structural variants comprise only a small fraction (0.14%) of the variants listed in Table 1.
- Repeats are scattered throughout the human genome, often embedded within both protein-coding and non-coding genes. Have the authors checked the fraction of exonic and intronic sequences dropped by masking repeats?
- Species classification results (100% accuracy) are somewhat concerning. Is the model potentially “cheating” by memorizing species-specific allelic changes?

---

### Official Review · Reviewer_EUqg · 2024-11-03

**Soundness:** 3
**Presentation:** 3
**Contribution:** 2
**Rating:** 5
**Confidence:** 4

**Summary:**

The paper introduces dnaGrinder, a genomic foundation model that utilizes recent developed methods in language modeling for efficient processing of long genomic sequences. The model integrates multiple known techniques, including Flash Attention 2, Sequence Length Warmup, and Attention with Linear Biases, to address challenges in computational efficiency. The paper also proposes a novel memory-efficient BPE tokenizer to improve memory usage. The authors evaluate dnaGrinder on several downstream genomic tasks and report comparable or slightly improved results over existing models, such as DNABERT-2 and Nucleotide Transformer.

**Strengths:**

1. The authors design a new data compression algorithm ME-BPE to address the memory usage issue when applying BPE on long DNA sequences.
2. The paper have a wide-range benchmarking across various downstream tasks.
3. The authors provide a detailed introduction of each technique and the experimental settings and results.

**Weaknesses:**

1. The work seems to lack substantial novelty on the method design. Since most modules incorporated are previously proposed (e.g. the Flash Attention, SLW and ALiBi).
2. While dnaGrinder demonstrates efficiency improvements, it performance on some tasks (e.g. Epigenetic Marks Prediction, Core Promoter Detection and Promoter Detection in Table 3) is not competitive enough.
3. Given the work relies on multiple components described in the paper, it's better to have an ablation study to illustrate the impact of each component.

**Questions:**

1. Because dnaGrinder doesn't achieve the best or second performance on certain tasks, could the authors discuss the trade-offs between efficiency and accuracy? Are there any conclusions about on which scenarios that dnaGrinder will work best?
2. Could the authors conduct ablation study on each components?
3. Since the ME-BPE is a novel design from the paper, how does it impact the performance improvement of dnaGrinder?

---

### Official Review · Reviewer_K36L · 2024-11-04

**Soundness:** 2
**Presentation:** 3
**Contribution:** 2
**Rating:** 5
**Confidence:** 4

**Summary:**

In this paper, the authors present a new DNA language model focused on improved efficiency. The model is shown to achieve SOTA performance on various downstream tasks while maintaining a small model size and low computational costs. Overall, I think this work falls short in terms of conceptual advancement but could be viewed as a good engineering effort. The benchmarking results show promising performance but critically lack ablation studies. Therefore I currently recommend a weak rejection of this manuscript.

**Strengths:**

1. The manuscript is overall well-written and easy to follow
2. The authors incorporated an array of engineering techniques and the resulting model is shown to be performant and efficient.
3. The evaluation, while not the most comprehensive, included most of the key downstream tasks as well as the prominent competing models.

**Weaknesses:**

Conceptually, the proposed points of novelty are mostly incremental advancements. Therefore, it is important to provide convincing empirical evidence. While the model shows promising performance on the benchmarks, it is unclear how much each modification contributes to the improvement. Ablation studies should be conducted to analyze the individual contributions of each component, which is vital to support the claims. More specifically:

a. Flash attention 2: This seems like a minor contribution, especially since DNABERT-2 already integrated flash attention.

b. Sequence length warm-up: HyenaDNA used this technique. Why is being the first encoder-only model to use it considered notable? While adapting it for variable-length BPE-tokenized sequences is a nice idea, an ablation study would be more convincing to show how much performance gain comes from this technique.

c. ME-BPE: This appears to be a minor modification of BPE, which is already used by DNABERT-2. There is also no assessment of how much memory this strategy actually saves.

d. Population data augmentation: it appears to be a small modification to the NT 1kG dataset. Though I acknowledge it to be a unique contribution. Again, it is unclear how much of the performance gain is attributed to this modification.

**Questions:**

1. The authors synthesized a list of tasks from a few existing benchmarks. I wonder why the authors left out the BEND benchmark (https://openreview.net/forum?id=uKB4cFNQFg), which I think is one of the most recognized and well-designed datasets in the field.

2. It would be helpful to add columns in Table 2 to describe context sizes.

3. Tables 3, 4, and 5 are very informative results and should be in the main text, or at least some summarized version. Just having one global average score in Table 2 does not provide sufficient detail.

4. In the species classification task, the performance of HyenaDNA is quite different from the their original results, where the model achieved Acc. 0.93 with a smaller context of 32k (Table 4.5 in the original paper). Also, the perfect performance by DNAGrinder is rather rare to see in genomics. Could the authors elaborate on the exact experimental setup?

---

### Official Review · Reviewer_Njzs · 2024-11-04

**Soundness:** 1
**Presentation:** 2
**Contribution:** 1
**Rating:** 3
**Confidence:** 5

**Summary:**

This paper introduces the model dnaGrinder, a lightweight encoder-only genomic foundation model. dnaGrinder is engineered to handle long sequences (140k tokens with 80GB GPU memory)  while using exact self-attention. The authors indicate the following major contributions:

- Using Flash Attention 2 for more efficient pretraining and inference
- Using Sequence Length Warmup (SLW) for more stabilized pretraining
- A memory-efficient BPE (ME-BPE) tokenizer training scheme
- An approach to remove repetitive regions to maintain the diversity of the pretraining dataset

On the downstream tasks from 1) the GUE benchmark from DNABERT-2 and 2) two enhancer-related tasks from the Nucleotide Transformer paper, dnaGrinder outperforms baselines such as NT-2500M-multi and DNABERT-2.

**Strengths:**

- dnaGrinder’s architecture is well-considered, with sensible design choices that balance efficiency and performance. The authors’ insights into pretraining and fine-tuning strategies add practical value for implementing genomic models.
- The model is benchmarked against well-known genomic language models like DNABERT-2 and Nucleotide Transformer, showing strong performance in accuracy and efficiency in the aforementioned benchmarks.
- The authors made commendable efforts in documenting dataset curation and preprocessing steps. This transparency is valuable for reproducibility.

**Weaknesses:**

The following points are my major concerns regarding the highlighted contributions in the paper:

* As a method development paper, the algorithmic and engineering innovations presented are minimal. While the authors position this work as introducing a novel lightweight genomic foundation model, the primary architectural differences from DNABERT-2 are:

   * Flash Attention 2 replacing Flash Attention Triton,
   * ALiBi positional encoding instead of RoPE,
   * and GEGLU activation in place of SwiGLU.

  These modifications are sensible engineering adjustments for language model development but do not constitute genuine innovations.

* While Sequence Length Warmup (SLW) has shown advantages in training decoder-only models by introducing progressively longer, more complex sequences akin to curriculum learning, its relevance for encoder-only genomic models is uncertain. Given that dnaGrinder’s tokenized sequences vary only moderately in length (700-2300 tokens, as reported), simpler alternatives—such as truncating sequences to a max_token_length or using balanced sampling (e.g., LengthGroupedSampler) for length-consistent batches—could likely achieve similar benefits.

* The introduction of ME-BPE in dnaGrinder seems aimed at solving a memory efficiency problem that may have limited relevance in practical genomic modeling. While training the BPE tokenizer can indeed be memory intensive, applying it during pretraining and inference requires minimal memory overhead. Even if a system lacks the memory to load the entire corpus, cloud vendors provide high-memory instances at reasonable costs, making this less of a barrier. Moreover, in NLP, tokenizers are often trained on a representative subset of the corpus (using options like --input_sentence_size=<size> and --shuffle_input_sentence=true in sentencepiece), rather than the full dataset used for language model training. This subsampling approach effectively reduces memory requirements without compromising vocabulary quality or requiring the additional complexity of ME-BPE’s iterative adjustments. As a result, ME-BPE seems to address a problem that is not particularly impactful in real-world scenarios.

* The authors claim that this work is the first to apply repetitive sequence removal, showing a clear disregard for the existing literature. Numerous models have already addressed repetitive elements using various techniques: down-weighting (GPN [1], PlantCaduceus [2]), downsampling/removal (SpliceBERT [3], GPN-MSA [4], hgT5 [5]), and fragment deduplication (gLM2 [6]). Notably, hgT5 also utilized RepeatMasker to identify repeats. The proposed method aligns with what should be considered standard practice, rather than an innovation.

Minor issues:
* The repetitive element removal method is introduced in the Human Reference Genome Dataset section, but the authors should clarify whether it was applied only to the human reference genome or to the entire combined dataset. If the latter is the case, this removal method would be better placed in the Methods section for consistency and clarity.
* Encoder-only models like dnaGrinder are non-causal and do not require KV caching for inference or embedding extraction. The authors' claim that Flash Attention 2’s inference optimization is "especially beneficial for our model...with most not exceeding 1,000 bp" is factually incorrect. In BERT-style models, inference is achieved through a single forward pass without the need for KV caching, which is specifically beneficial for autoregressive generation—a context irrelevant to dnaGrinder. Thus, the supposed inference advantage of Flash Attention 2 does not apply here.

* Some suggested improvements to the paper:
   - In Figure 1a, the plot displays two masked token embeddings, yet the tokens in “ME-BPE Tokenization” are not masked. Aligning these would improve clarity.
   - In the last paragraph of Section 2.2.2, Further Pretraining, the phrase “our model performed 100,000 steps” should be corrected to “DNABERT-2 model performed 100,000 steps.”
   - DNABERT-2 should be recognized as the state-of-the-art model in the benchmark alongside dnaGrinder, as it not only holds the highest combined number of top-1 and top-2 rankings but also has the highest average scores.
   - In the results table, the “Params” header is marked with a ↓ symbol, explained as “↓ indicates that a lower value is better.” However, the question of whether smaller or larger language models perform better for genomic modeling remains unresolved. It would be beneficial for the authors to discuss this open question.

References:
1. Benegas, G., Batra, S. S., & Song, Y. S. (2023). DNA language models are powerful predictors of genome-wide variant effects. Proceedings of the National Academy of Sciences, 120(44), e2311219120.
2. Zhai, J., Gokaslan, A., Schiff, Y., Berthel, A., Liu, Z. Y., Miller, Z. R., ... & Kuleshov, V. (2024). Cross-species modeling of plant genomes at single nucleotide resolution using a pre-trained DNA language model. bioRxiv, 2024-06.
3. Chen, K., Zhou, Y., Ding, M., Wang, Y., Ren, Z., & Yang, Y. (2024). Self-supervised learning on millions of primary RNA sequences from 72 vertebrates improves sequence-based RNA splicing prediction. Briefings in Bioinformatics, 25(3), bbae163.
4. Benegas, G., Albors, C., Aw, A. J., Ye, C., & Song, Y. S. (2023). GPN-MSA: an alignment-based DNA language model for genome-wide variant effect prediction. bioRxiv.
5. Ioannidis, N. (2024, March). GUANinE v1. 0: Benchmark Datasets for Genomic AI Sequence-to-Function Models. In Machine Learning in Computational Biology (pp. 250-266). PMLR.
6. Cornman, A., West-Roberts, J., Camargo, A. P., Roux, S., Beracochea, M., Mirdita, M., ... & Hwang, Y. (2024). The OMG dataset: An Open MetaGenomic corpus for mixed-modality genomic language modeling. bioRxiv, 2024-08.
7. Marin, F. I., Teufel, F., Horlacher, M., Madsen, D., Pultz, D., Winther, O., & Boomsma, W. (2023, November). Bend: Benchmarking dna language models on biologically meaningful tasks. In The Twelfth International Conference on Learning Representations.

**Questions:**

* The authors also did not include any ablation studies or analyses to verify SLW’s effectiveness in this setting, leaving its impact and necessity unclear.
* The authors benchmarked dnaGrinder solely against other genomic language models, omitting conventional genomic baseline methods. Without these established baselines, it remains unclear whether the genomic LMs are themselves underperforming, making it challenging to assess dnaGrinder’s effectiveness fully. The paper would be strengthened by including newer benchmarks like BEND [7], which incorporates traditional baselines, and/or variant effect prediction tasks on ClinVar, particularly relevant since the model is trained on the human genome and VCF data.

---

### Note · Authors · 2024-11-26

I have read and agree with the venue's withdrawal policy on behalf of myself and my co-authors.